# Immunotherapy of Multiple Myeloma: Current Status as Prologue to the Future

**DOI:** 10.3390/ijms242115674

**Published:** 2023-10-27

**Authors:** Hanley N. Abramson

**Affiliations:** Department of Pharmaceutical Sciences, Eugene Applebaum College of Pharmacy and Health Sciences, Wayne State University, Detroit, MI 48202, USA; ac2531@wayne.edu

**Keywords:** myeloma, daratumumab, bispecific antibodies, chimeric antigen receptor T-cells, immunotherapy

## Abstract

The landscape of therapeutic measures to treat multiple myeloma has undergone a seismic shift since the dawn of the current century. This has been driven largely by the introduction of new classes of small molecules, such as proteasome blockers (e.g., bortezomib) and immunomodulators (e.g., lenalidomide), as well as by immunotherapeutic agents starting with the anti-CD38 monoclonal antibody daratumumab in 2015. Recently, other immunotherapies have been added to the armamentarium of drugs available to fight this malignancy. These include the bispecifics teclistamab, talquetamab, and elranatamab, and the chimeric antigen receptor (CAR) T-cell products idecabtagene vicleucel (ide-cel) and ciltacabtagene autoleucel (cilta-cel). While the accumulated benefits of these newer agents have resulted in a more than doubling of the disease’s five-year survival rate to nearly 60% and improved quality of life, the disease remains incurable, as patients become refractory to the drugs and experience relapse. This review covers the current scope of antimyeloma immunotherapeutic agents, both those in clinical use and in development. Included in the discussion are additional monoclonal antibodies (mAbs), antibody–drug conjugates (ADCs), bi- and multitargeted mAbs, and CAR T-cells and emerging natural killer (NK) cells, including products intended for “off-the-shelf” (allogeneic) applications. Emphasis is placed on the benefits of each along with the challenges that need to be surmounted if MM is to be cured.

## 1. Introduction

Multiple myeloma (MM) is a hematologic cancer characterized by a clonal proliferation of plasma cells in the bone marrow, as well as by high levels of monoclonal immunoglobulins in blood and/or urine [1]. As a blood cancer, the disease ranks 2nd in the United States behind non-Hodgkin’s lymphoma (NHL) and 14th among all cancers in terms of incidence. Based on current estimates, in 2023, MM will be diagnosed in 35,730 individuals (55.6% male) and will be responsible for 12,590 deaths in the U.S. [2]. The median age at diagnosis in the U.S. is 69 [3]. Worldwide, in 2020, there were an estimated 176,404 cases of the disease (0.9% of all cancers) accounting for 117,077 deaths (1.2% of cancer deaths) [4]. Substantial racial disparities have been noted for all stages of MM in the U.S. For example, in the 2014–2018 period, the incidence of MM per 100,000 population in African Americans was more than double that in Caucasians (16.7 vs. 7.8 for males and 12.3 vs. 4.8 for females) [5]. Guidelines for the diagnosis and treatment of MM are updated and published annually by the National Comprehensive Cancer Network (NCCN) [6]. While the cause of MM is not established, cytogenetic factors are known to play a significant role in certain MM patients classified as “high risk”. Among the most frequently observed of these variances are the chromosomal deletion del(17p) and the transversions t(4;14) and t(14;16) [7]. 

An active case of MM typically includes a quartet of symptoms known by the acronym CRAB: hyper***c***alcemia, ***r***enal insufficiency, ***a***nemia, and ***b***one lesions. An asymptomatic state, monoclonal gammopathy of undetermined significance (MGUS) often precedes an active case [8]. The risk of progression from MGUS to MM has been estimated at about 1% per year [9]. Smoldering multiple myeloma (SMM), a second asymptomatic phase, has also been described, although its significance relative to MM has been a matter of continuing controversy [10].

Alkylating agents (melphalan and/or cyclophosphamide), often in combination with corticosteroids, were the treatment of choice for MM starting in the 1950s. This standard regimen was augmented by autologous stem cell transplantation (ASCT) in the mid-1980s. The picture began to change in the late 1990s with the discovery of the antimyeloma effects of thalidomide and later, of its two chemically related derivatives, lenalidomide and pomalidomide. Furthermore, the first two decades of the current century were marked by the discovery of the antimyeloma benefits of two entirely new classes of small molecules—proteasome inhibitors (bortezomib, carfilzomib, and ixazomib) and blockers of nuclear export (selinexor). 

A paradigm shift in the therapeutic approach to MM occurred in 2015 with the approval of two monoclonal antibodies, daratumumab and elotuzumab, for the disease. This review covers the fundamental aspects of MM immunotherapy, highlighting the remarkable advances made in this arena over the past decade and features a discussion of currently approved agents, as well as those in various stages of development. For information on the background, history, and current status of immunotherapeutic applications in cancer treatment, the reader is directed to the recent reviews of Liu [11] and Anagnostou [12]. 

## 2. Monoclonal Antibodies Targeting CD38

The 45 kDa transmembrane glycoprotein CD38 has elicited considerable interest as a major target in MM. This biomarker, which is expressed at high levels in both normal and neoplastic plasma cells, as well as at lower levels in other blood cells, is known to perform several functions in the cell. Its primary role appears to result from its cyclic ADP ribose hydrolase activity, the products of which are important regulators of intracellular calcium levels. The lethality of anti-CD38 antibodies to myeloma cells is due to a combination of antibody-dependent cellular cytotoxicity (ADCC), antibody-dependent cellular phagocytosis (ADCP), and complement-dependent cytotoxicity (CDC) [13]. 

Since its approval by the U.S. Food and Drug Administration (FDA) as the first immunotherapeutic agent for MM, daratumumab (Darzalex^®^ by Janssen, Beerse, Belgium), a fully human IgG1κ monoclonal antibody (mAb) that targets CD38^+^ cells, has assumed a prominent role in the treatment of the disease. According to NCCN guidelines, the combination of daratumumab with lenalidomide, bortezomib, and dexamethasone is considered a major recommendation in newly diagnosed multiple myeloma (NDMM) for patients regardless of transplant eligibility status. Additionally, daratumumab–dexamethasone triplet regimens that include either lenalidomide or bortezomib are primary recommendations in cases of refractory and/or relapsed multiple myeloma (RRMM). A number of network meta-analytic studies of random-controlled trials may be cited as recent examples that serve to support the application of daratumumab as a major player in a number of myeloma-based settings [14,15,16,17,18]. Moreover, a subcutaneous formulation of daratumumab/hyaluronidase has recently become available. This innovation, which enables a shorter administration period—3 to 5 min as opposed to the several hours required for intravenous (iv) infusion—has been accomplished without sacrificing therapeutic efficacy or patient safety [19]. HLX15, a daratumumab biosimilar developed in China recently entered a phase II clinical trial designed to compare its properties with those of daratumumab in normal male subjects (NCT05679258). Infusion reactions, which include rash, headache, cough, nausea, vomiting, nasal congestion, and dyspnea, represent an adverse but usually low-grade reaction to daratumumab, being noted in up to 50% of patients receiving the drug, especially during the first two infusions. Prior administration of a glucocorticoid and/or a leukotriene blocker (e.g., montelukast) may help diminish this effect and may be especially useful in patients with an underlying respiratory disease [20]. In addition, daratumumab can interfere with blood typing by binding to CD38 on reagent blood cells, resulting in a positive indirect Coombs test. This can be circumvented by performing blood-typing procedures prior to using daratumumab [21]. Daratumumab use may also result in infections primarily resulting from bone marrow suppression. This may necessitate the use of prophylactic antimicrobials such as cotrimoxazole to prevent Pneumocystis jiroveci pneumonia. 

Isatuximab, (Sarclisa^®^ by Sanofi, Paris, France), a second anti-CD38 mAb, was introduced for MM in 2020. The mechanism of action of this chimeric IgG1κ mouse–human construct is similar to that of daratumumab, although isatuximab binds to an epitope distinct from that which binds daratumumab [22]. Isatuximab was approved initially for intravenous use with pomalidomide and dexamethasone in patients who had relapsed following at least two prior therapies that included a proteasome inhibitor and lenalidomide [23]. Data from the phase III ICARIA trial, in which the addition of isatuximab to a pomalidomide/dexamethasone regimen in RRMM subjects resulted in significantly longer progression-free survival (PFS) (11.5 months vs. 6.5 months), served as basis for this approval [24]. Subsequently, the replacement of the immunomodulator in this triplet therapy by the proteasome blocker carfilzomib was sanctioned for use in RRMM patients who had received as few as one to three prior regimens. Approval in the latter case was based on the PFS, duration of response, and safety data provided by the phase III IKEMA trial (NCT03275285) [25,26]. Infusion-related reactions, fatigue, hypertension, diarrhea, and upper respiratory tract infections are among the most common adverse reactions noted with isatuximab. The efficacy of subcutaneous administration of isatuximab in regimens containing pomalidomide and dexamethasone is currently under study in two clinical trials: NCT04045795 (phase I, N = 56) [27,28,29] and NCT05405166 (phase III, N = 534). In addition, a recently initiated phase II trial (NCT05704049; N = 68) is investigating the potential role of subcutaneous isatuximab in a carfilzomib/dexamethasone regimen in RRMM patients. 

## 3. Monoclonal Antibodies Targeting SLAMF7

The signaling lymphocytic activation molecule family (SLAMF) comprises a group of transmembrane glycoproteins that is highly expressed almost exclusively on the surface of plasma cells from both normal and MM patients, as well as on natural killer (NK) cells. One member of this family, SLAMF7 (CS1), has emerged as a major target in the search for new immunological products with antimyeloma activity [30]. The most promising of these agents, the anti-SLAMF7 humanized IgG1κ mAb elotuzumab (Empliciti^®^ by Bristol Myers Squibb, Princeton, NJ, USA), was approved for use in combination with lenalidomide–dexamethasone in RRMM patients who had relapsed following one to three prior therapies. Elotuzumab, which, unlike anti-CD38 drugs, lacks single-agent activity, works through the direct activation of SLAMF7 on both myeloma cells and NK cells to kill myeloma cells via ADCC. Moreover, the direct binding of elotuzumab’s Fc portion to the FcγRIII (CD16) receptor on NK cells releases perforins and granzymes, which are cytotoxic to myeloma cells [31,32]. The approval of elotuzumab followed an analysis of the results from the ELOQUENT-2 trial (NCT01239797) that included 646 patients who were randomly assigned to receive elotuzumab–dexamethasone (Ed) with or without lenalidomide (R). A 30% reduction in the risk of death or disease progression was found for the ERd cohort after one year [33]. Subsequent multiyear follow-up data confirmed the efficacy of this triplet regimen for RRMM [34,35,36]. Further evidence of the benefits of combining Ed treatment with an immunomodulator was provided by the ELOQUENT-3 trial (NCT02654132), which included pomalidomide in patients refractory to both lenalidomide and a proteasome inhibitor [37]. Accordingly, the elotuzumab–dexamethasone–pomalidomide regimen received FDA sanction in 2018 for the treatment of RRMM patients who had received at least two prior therapies that included these two agents. No increased safety concerns were identified in any of the studies in which elotuzumab was combined with immunomodulators. Trials combining elotuzumab–dexamethasone with the proteasome inhibitors bortezomib [38] or carfilzomib [39] in RRMM, although demonstrating significant efficacy, to date have failed to reach the level of favorable outcomes of the scale attained with elotuzumab–immunomodulator combinations. Initial efforts aimed at developing a subcutaneous dosage form of elotuzumab have been described [40].

## 4. Antibody–Drug Conjugates

A number of antibody–drug conjugates (ADCs), drugs in which an antibody is covalently linked to a toxin or radionuclide, have emerged in the past decade to play an important role in the therapy of certain types of cancer, such as breast and cervical cancers, acute leukemias, and Hodgkin’s disease and other types of lymphoma. ADC construction is predicated on three components: (1) an antibody which targets a tumor-specific or tumor-associated antigen (TSA or TAA), (2) a small cytotoxic molecule (the payload), and (3) a linker to attach the payload to the antibody. Endocytosis of the ADC and subsequent lysosomal cleavage releases the payload to initiate apoptosis of the cancer cell. The antibodies used are generally of the IgG1 class since they have demonstrated a potent capacity to activate ADCC and ADCP by specific FcγRIIIa binding to NK cells [41]. This binding is further enhanced by the removal of fucosyl groups (afucosylation) from the N-linked biantennary complex oligosaccharides in the antibody’s Fc region [42]. The linkers used may be either cleavable or noncleavable. The former includes chemical functionalities such as hydrazones, disulfides, and peptides, which employ a specific enzymatic property inherent to the tumor cell, while the latter group depends on a complete lysosomal degradation of the antibody to cause the release of both payload and linker [43,44].

B-cell maturation antigen (BCMA; CD269; TNFRSF17), a 184-amino acid transmembrane glycoprotein belonging to the tumor necrosis family (TNF), has emerged recently as a highly attractive TAA in the quest for new agents to treat MM. This interest is driven by reports that both BCMA and its mRNA transcript are almost exclusively confined to plasma cells and are overexpressed consistently during the malignant transformation to myeloma in both patient samples and cell lines [45]. Moreover, there is much evidence indicating that membrane-bound BCMA is a reliable biomarker for MM diagnosis and prognosis, as well as a predictor of response to treatment [46]. On the other hand, although BCMA acts as a ligand for BAFF (B-cell activating factor) and APRIL (a proliferation-inducing ligand), inhibitors of these two cytokines, which play key roles in myeloma cell proliferation and viability, have demonstrated poor results in MM trials [47,48,49].

The extracellular domain of membrane-bound BCMA may be cleaved by γ-secretase resulting in the release of soluble BCMA (sBCMA) into the plasma, high levels of which have been correlated with poor clinical outcomes in MM patients. sBCMA not only is an indicator of reduced density of the membrane-bound target antigen BCMA but also serves as a soluble decoy to reduce the efficacy of anti-BCMA agents [50,51]. It is for this reason that orally effective γ-secretase inhibitors, such as nirogacestat and crenigacestat, have been included in clinical trials of some newer anti-BCMA drugs [52]. 

The first and thus far only ADC approved for MM is belantamab mafodotin (Blenrep^®^ by GSK, Brentford, UK), an afucosylated humanized BCMA-targeted antibody that is coupled to the tubular polymerization blocker monomethyl auristatin F (MMAF, mafodotin) by a noncleavable maleimidocaproyl linker. Initial FDA approval in 2020 of belantamab mafodotin was provided on an accelerated basis for use as monotherapy in RRMM patients who had received at least four earlier regimens, including an anti-CD38 mAb, a proteasome inhibitor, and an immunomodulator [53,54]. Approval was based on data from the DREAMM-2 trial (NCT03525678), which included a total of 196 RRMM patients receiving the drug as a single agent divided into two dosage groups: 2.5 mg/kg (N = 97) and 3.4 mg/kg (N = 99). Overall response rate (ORR) and PFS data recorded for the two cohorts were 31% and 2.9 months and 34% and 4.9 months, respectively [55]. However, the FDA-required follow-up confirmatory trial (DREAMM-3; NCT04162210) in which belantamab mafodotin monotherapy was compared against pomalidomide/dexamethasone failed to demonstrate any improvement in PFS and the approval was withdrawn in November 2022 [56]. Nevertheless, a number of trials in the DREAMM (DRiving Excellence in Approaches to Multiple Myeloma) series that include comparisons of belantamab mafodotin with established antimyeloma therapies remain active (see Table 1). 

Blood cell dyscrasias, such as anemia and thrombocytopenia, were among the most commonly noted adverse effects associated with the use of belantamab mafodotin in the DREAMM-2 trial, as well as in the precursor phase I study. However, the most serious toxic effect accompanying the use of this ADC has been the corneal microcyst-like epithelial changes (MECs) first reported in 72% (68/95) of DREAMM-2 patients [76]. This effect, which is not subject to corticosteroid relief but may be reversed by dose reduction [77], has been consistently observed in other studies of not only this but other MMAF-containing ADCs, although its mechanism is unknown [78,79,80,81]. Consequently, the initial FDA approval of belantamab mafodotin was accompanied by a boxed-warning requirement, stating that changes in the corneal epithelium may result in vision loss, corneal ulcers, blurred vision, and dry eyes. As a result, the drug is available only under a restricted Risk Evaluation and Mitigation Strategy (REMS) program, which remains applicable to patients receiving the drug under the provisions of compassionate use. 

α-Amanitin, a highly toxic bicyclic octapeptide produced by *Amanita phalloides* (“death cap mushroom”) has recently emerged as a cytotoxin exhibiting a novel mode of action among ADCs, viz, the inhibition of RNA polymerase II [82]. HDP-101, an ADC consisting of an α-amanitin analog conjugated to a BCMA-targeted antibody by a cathepsin B-cleavable linker, is currently the subject of a phase I safety-assessment trial (NCT04879043) in RRMM patients. Initial results on the first cohort of patients showed the drug to have good tolerability in late-stage disease [83]. 

Another BCMA-targeted ADC of current interest is AMG 224, an afucosylated IgG1 mAb coupled to the microtubule blocker mertansine through a maleimidocaproyl non-cleavable linker. An initial report from a phase I trial (NCT02561962) of AMG 224 showed an ORR of 23% in 40 heavily pretreated RRMM subjects [84]. CC-99712, bearing a maytansinoid payload through a noncleavable dibenzocyclooctyne linker, is another BCMA-targeted ADC now included in an active myeloma-based trial that also incorporates a γ-secretase inhibitor (NCT04036461). However, no data have yet been reported for this study. Besides BCMA, a number of other myeloma-associated targets have served as the basis for the formulation of novel ADCs now in clinical development. These are shown in Table 2. In addition, clinical trials of some ADCs that once appeared promising for RRMM have been halted for a variety of reasons, such as poor activity, unacceptable toxicity, and/or undisclosed sponsor decision factors. These include (target in parentheses) MEDI2228 (BCMA), azintuxizumab vedotin (SLAMF7), lorvotuzumab mertansine (CD56), and DFRF4539A (FcRH5). 

## 5. Bispecific Antibodies

T-cell-based immunotherapeutic approaches to cancer have risen to a position of great prominence in recent years. In particular, two specific areas of research have dominated this field: T-cell-engaging bispecific antibodies (T-BsAbs) and chimeric antigen receptor (CAR) T-cell therapies. The former is described in this section while discussion of the latter is reserved for the following segment. 

Predicated on a concept originally proposed in 1961 by Nisonoff [93], T-BsAbs are based on the construction of a dual-targeting antibody in which one arm initially attaches to the T-cell CD3 coreceptor, enabling the subsequent binding of a second arm to a TSA or TAA on the targeted cancer cell. Although the concept has a multitude of variations, the basic strategy whereby cytotoxic T-cells become tethered to tumor cells results in the latter’s cytolysis. Cell killing is attributed to the combined effects of two proteins: perforin, which produces transmembrane pores in tumor cells, and granzyme B, which travels through the thus-formed channels to activate apoptosis [94].

Bispecific antibodies provide certain theoretical benefits by circumventing certain processes normally associated with immune responses, such as costimulatory molecules, antigen-presenting cells, or the interplay between antigens and major histocompatibility complexes (MHC). In addition, the polyclonal expansion of T memory cells is enabled by persistent T-cell activation. Furthermore, “off-the-shelf” products are made possible through a modification of the relative affinity of each arm for its specific target, thus permitting the tweaking of each construct’s therapeutically relevant properties to optimize biopharmaceutic parameters and activity [93,94,95,96]. The Bi-specific T-cell Engager (BiTE^®^) platform represents an application of the bispecific concept in which two single-chain variable fragments (scFvs) arranged in tandem provide the cross-link between the T-cell CD3 coreceptor and a tumor cell TSA or TAA. Originally developed by Micromet AG, (Munich, Germany, now a subsidiary of Amgen), BiTE^®^ technology was first successfully applied to cancer therapy with the approval in 2014 of blinatumomab for B-cell precursor acute lymphocytic leukemia (B-cell ALL). The scFv cross-link in this product is provided between T-cell CD3 and tumor cell lymphocyte antigen CD19 although studies seeking to extend this model to myeloma cell CD19 as a suitable binding partner have proven much less successful. On the other hand, BCMA has demonstrated itself to be most worthy of exploration as a CD3 linkage companion. Success in this space was attained with FDA’s authorization of the IgG4-based teclistamab (Tecvayli^®^ by Janssen) in October 2022 as the first CD3 x BCMA bispecific for RRMM. Approval, which was granted for subcutaneous use in patients who had received at least four lines of therapy, was based on the evaluation of data from the MajesTEC-1 trial (NCT03145181; NCT04557098). That study revealed an ORR of 63.0% (104/165) and a median PFS of 11.3 months in the 128 (77.6%) subjects classed as triple-class refractory. Notably, 44 (26.7%) of the evaluated patients gave no evidence of minimal residual disease (MRD). The most common adverse effects observed were cytokine release syndrome (CRS) (72.1%), neutropenia (70.9%), anemia (52.1%), thrombocytopenia (40.0%), and neurotoxicity (14.5%). In addition, infections were noted in 76.4% of the patients [95]. Several of the currently active trials of teclistamab are shown in Table 3.

In August 2023, the IgG2a construct elranatamab (Elrexfio^®^ by Pfizer, New York, NY, USA), another CD3 x BCMA bispecific that has emerged in this class, was granted accelerated approval as a monotherapy by the FDA [107]. This action was spurred by the results from the MagnetisMM-3 trial (NCT04649359) in which an ORR of 61.0% and a complete response rate of 35.0% were reported in a cohort of 123 RRMM subjects after a median follow-up of 14.7 months [108]. The most frequently observed grade 3 or higher adverse events in this trial were infections (39.8%), anemia (37.4%), and neutropenia (48.8%). CRS of any grade was found to occur in 57.7% but no patients experienced CRS higher than grade 2. Overall, biweekly dosing was found to reduce total grade 3 or 4 adverse events from 58.6% to 46.6%, indicating that biweekly dosing may offer the advantage of improved safety without compromising efficacy [108]. Table 4 contains a listing of several ongoing clinical trials in which elranatamab is under evaluation. 

An alternative strategy for developing bispecifics targeting BCMA is represented by receptors on NK cells, which, like cytotoxic T-cells, release granzyme and perforin, as well as certain apoptosis-inducing ligands [115]. A prominent example here is RO7297089 (AFM26), which engages myeloma cells BCMA and NK CD16A and is currently the subject of a phase I trial for RRMM (NCT04434469). An initial report on 27 patients found the drug to be well tolerated although efficacy was weak with only four subjects evidencing partial or minimal responses [116]. 

In addition to the bispecifics noted above, several other variations based on the CD3 x BCMA theme are now under development. These include (see Table 5 for clinical trials incorporating these and related agents):Linvoseltamab (REGN5458): an IgG4κ-based CD3 x BCMA bispecific developed using Regeneron’s VelocImmune^®^ “human antibody mouse” technology [117]. This product differs from the closely related REGN5459 primarily in its binding characteristics with the CD3 arm [118].EMB-06: a tetravalent CD3 x BCMA bispecific antibody in which two antigen-binding fragments are fused in a crisscross fashion, referred to as a Fabs-In-Tandem-Ig format, to generate four independently active antigen binding sites [119].Alnuctamab (CC-93269): a 2 + 1 T-cell engager from Bristol Myers Squibb with two arms binding to BCMA and one arm to CD3ε [120].HPN217: a trispecific engager with a prolonged half-life developed by Harpoon Therapeutics in which the three binding domains are combined in a single chain—an N-terminal BCMA-binding portion, a C-terminal CD3e T-cell binding component, and a central human serum albumin-binding domain [121].ABBV-383 (TNB-383B): the product of a joint venture between AbbVie and Tenebio, containing two variable heavy chains and a single light chain in a BCMA x CD3ε format [122].WVT078: a CD3 x BCMA IgG1 construct with an IgG1 backbone containing Fc mutations designed to favor the pairing of the two distinct heavy chains [123].

**Table 5 ijms-24-15674-t005:** Selected active trials of additional CD3 x BCMA targeting bispecific antibodies in multiple myeloma (MM).

Trial ID (References)	Drugs	Phase	Enrollment (N)	Trial Title
NCT05650632	ABBV-383	I	80	Multicenter, Open-label Study to Evaluate Dose Optimization Measures and Safety of ABBV-383 in Subjects with RRMM
NCT05259839 [124]	ABBV-383 + various antimyeloma agents	I	270	Dose Escalation and Expansion Study of ABBV-383 in Combination with Anti-Cancer Regimens for the Treatment of Patients with RRMM
NCT04735575	EMB-06	I/II	66	First-in-human, Open-Label Study to Evaluate the Safety, Tolerability, Pharmacokinetics and Preliminary Antitumor Activity of EMB-06 in Patients with RRMM
NCT03486067 [125]	Alnuctamab (CC-93269)	I	220	Open-label, Dose Finding Study of CC-93269, a BCMA x CD3 T Cell Engaging Antibody, in Subjects with RRMM
NCT05828511	Linvo (REGN5458)	I/II	132	Study of Linvoseltamab (Anti-BCMA X Anti-CD3 Bispecific Antibody) in Previously Untreated Patients with Symptomatic MM
NCT05730036	Elo+ Pom + Dex +/−	III	286	Open-label, Randomized Study of Linvoseltamab (REGN5458; Anti- BCMA x Anti-CD3 Bispecific Antibody) Versus the Combination of Elotuzumab, Pomalidomide, and Dexamethasone (EPd), in Patients with RRMM (LINKER-MM3)
NCT05137054	Linvo + various anti-myeloma agents	Ib	317	Study of REGN5458 (Anti-BCMA x Anti-CD3 Bispecific Antibody) Plus Other Cancer Treatments for Patients with RRMM
NCT03761108 [126,127,128]	Linvo	I/II	309	FIH Study of REGN5458 (Anti-BCMA x Anti-CD3 Bispecific Antibody) in Patients with RRMM
NCT04083534 [118]	REGN5459	I/II	43	FIH Study of REGN5459 (Anti-BCMA x Anti-CD3 Bispecific Antibody) in Patients with RRMM
NCT03933735 [129]	TNB-383B	I/II	220	Multicenter Open-label, Dose-escalation and Expansion Study of TNB-383B, a Bispecific Antibody Targeting BCMA in Subjects with RRMM
NCT05646758	TQB2934	I	140	Clinical Trial Evaluating the Tolerance and Pharmacokinetics of TQB2934 for Injection in MM Subjects
NCT04123418 [123]	WVT078	I	56	Open-label, Multicenter, Study of WVT078 in Subjects with RRMM
NCT04184050 [130]	HPN217	I	70	Open-label, Multicenter, Dose Escalation Study of the Safety, Tolerability, and Pharmacokinetics of HPN217 in Patients with RRMM

Dex = dexamethasone; Elo = elotuzumab; FIH = first in humans; Linvo = linvoseltamab; MM = multiple myeloma; Pom = pomalidomide; RRMM = relapsed and/or refractory MM.

Surface antigens other than BCMA have also served as targets for the discovery of antimyeloma bispecific antibodies. One that has risen to prominence in recent years is GPRC5D (G-protein-coupled receptor, class C, group 5, member D). This orphan receptor is known to be expressed primarily on the surface of myeloma cells, as well as on hard-keratinized tissues [131], and is associated with poor prognosis in MM [132]. In August 2023, the FDA granted accelerated approval to talquetamab (Talvey^®^ by Janssen), a GPRC5D x CD3 bispecific [133], based on the results of the MonumenTAL-1 trial (NCT03399799, NCT4634552) in which ORRs of 73.0% and 73.6% were obtained in respective cohorts receiving subcutaneous doses of either 0.4 mg/kg weekly (N = 100) or 0.8 mg/kg biweekly (N = 87). The most common adverse events attributed to talquetamab were mainly low grade and consisted of CRS, altered taste, and were skin-related. Table 6 shows the current ongoing trials of talquetamab in RRMM patients. 

Forimtamig (RG6234; RO7425781), a second CD3 x GPRC5D bispecific that has shown potential antimyeloma efficacy in preclinical studies [142] recently was advanced to a clinical study for the disease (NCT04557150). Initial results from a phase I dose-escalation investigation demonstrated that iv administration of this agent was somewhat more effective than sc use [iv: ORR = 71.4% (35/49) vs. sc: 60.4% (29/48)] [143]. 

Another non-BCMA target is FcRH5, which is expressed on virtually all malignant plasma cells [144]. The major entrant in this area is cevostamab (BFCR4350A; RO7187797), designed to bind T-cell CD3 with the most membrane-proximal domain of FcRH5 and currently the subject of six clinical trials for RRMM (see Table 7). It is noteworthy that each of these trials incorporates tocilizumab, which has been demonstrated to reduce cytokine release syndrome when given prior to cevostamab [145]. 

Included among other non-BCMA-directed constructs presently under consideration for RRMM are:ISB 1342 (GBR 1342), a CD3 x CD38 bispecific (NCT03309111) [149,150].VP301, a CD38 x ICAM-1 bispecific (NCT05698888); ICAM-1 (intercellular adhesion molecule-1; CD54), an adhesion molecule highly expressed on transformed plasma cells and associated with antimyeloma drug resistance [151,152].ISB 1442 (NCT05427812), a trispecific which engages both CD38 and CD47 on myeloma cells [153,154].SAR442257, another trispecific (CD38/CD28 x CD3) (NCT04401020) [155].

A recent systematic analysis of 18 RRMM-based studies (1283 total patients) involving bispecific antibodies found that the 13 studies that included BCMA-targeting agents had ORRs ranging from 25% to 100% while the 5 studies that involved non-BCMA-targeting agents had ORRs that ranged between 60% and 100%. Also, a comparison of rates of complete responses (or stringent complete responses) tended to favor bispecifics directed against targets other than BCMA: 19–63% vs. 7–38% [156]. 

## 6. CAR T-Cells

The past decade has witnessed the emergence of chimeric antigen receptor (CAR) T-cell therapy as an immunotherapeutic approach of immense importance in the treatment of various cancers [157,158]. As a mode of adoptive cell transfer (ACT), this strategy employs recombinant DNA technology using a viral vector to genetically modify a patient’s own cytotoxic T-cells, enabling the eventual expression of a chimeric receptor specifically targeting a TAA or TSA almost exclusively confined to a given type of malignant cell. CAR T-cells typically consist of the target-directed ectodomain (usually an scFv), a transmembrane component, a hinge region, and costimulatory signaling endodomains, such as CD28 or 4-1BB [159]. The thusly engineered T-cells then can be reinfused back into the patient resulting in T-cell engagement with the targeted cancer cells resulting in their death. This tactic has been most successfully used in hematological cancers, especially B-cell malignancies where CD19 is the intended target although, as noted earlier, myeloma cell CD19 per se has not been demonstrated to be particularly amenable as an immunotherapeutic target. For example, a study of tisagenlecleucel, an anti-CD19 CAR, combined with ASCT and melphalan, provided poor clinical benefit to a group of ten MM subjects (NCT02135406) [160]. However, an exception to this may be seen with CARs that target both CD19 and BCMA—see the discussion of GC012F (below). 

Some early trials involving anti-CD19 CAR T-cell therapy were confounded by the finding that durable remissions were difficult to obtain in spite of initial ORRs often exceeding 80% [161,162,163,164]. This issue was resolved eventually with the benefits of including lymphodepletion, usually consisting of combined fludarabine/cyclophosphamide, in the pre-ACT regimen [165]. The mechanism underlying the enhanced T-cell expansion and persistence, as well as the more favorable clinical outcomes, associated with lymphodepletion, remain obscure. 

In common with several of the drugs discussed above, BCMA also plays a concentrated role as a TAA of intense interest in antimyeloma CAR T-cell research. The first CAR T-cell product to receive FDA approval for MM (March 2021) was idecabtagene vicleucel (Abecma^®^ by Bristol Myers Squibb, ide-cel) [166]. This anti-BCMA construct employed a lentivirus-engineered template in which the anti-BCMA scFv was linked serially to a CD8 hinge, a transmembrane region, a CD28 costimulatory domain, and CD3ζ served as the T-cell activator. Approval focused on the results of the KarMMa-1 (NCT03361748) trial that included 128 RRMM patients who had failed at least three prior standard regimens. Of the 128 patients receiving the drug, a 73% ORR was seen and 42 (33%) experienced complete responses or better. A median PFS of 8.8 months was recorded. CRS was reported in 84% of subjects (5% of grades 3 or higher) while neurotoxicity was observed in 16% (3% grade 3; 0% grade 4 or higher) [167]. Further encouraging outcomes were seen in a recent RRMM phase III study (NCT03651128; KarMMa-3) that compared two cohorts of RRMM patients (2–4 prior regimens), one of which received ide-cel (N = 254) while the other (N = 132) was given one of five standard MM regimens. After 18.6 months of follow-up, a median PFS of 13.3 months was observed in the ide-cel group vs. 4.4 months in the group receiving standard courses of therapy. Toxicity in the ide-cel cohort was similar to that seen in the KarMMa-1 study [168]. Ide-cel’s sponsor (Bristol Myers Squibb) recently filed applications with the FDA and European Medicines Agency (EMA), as well as in Japan, to enable the agent to be used in triple-, instead of the originally sanctioned quadruple-, class-exposed RRMM patients [169]. Action on these applications remains pending as of this writing. 

In February 2022, the FDA approved a second BCMA-targeted CAR T-cell therapeutic drug for RRMM—ciltacabtagene autoleucel (Carvykti^®^, by Janssen and Legend, cilta-cel). Approval was supported by the results of the CARTITUDE-1 trial (NCT03548207) in which 97 patients with at least three prior lines of therapy achieved an ORR of 97% and 67% stringent complete responses. After 12 months, PFS and overall survival (OS) rates of 77% and 89%, respectively, were reported. CRS was noted in 95% of the study subjects (4% grade 3 or higher) while neurotoxicity was observed in 21% [170]. The results of a phase III trial (NCT04181827; CARTITUDE-4) that studied lenalidomide-refractory patients provided further evidence of the effectiveness of cilta-cel in RRMM patients [171]. That study compared one cohort (N = 208) which received a single injection of cilta-cel with a second group (N = 211) which received only standard care. The PFS of the first cohort after 12 months was 75.9% vs. 48.6% in the second group, while the ORRs were 84.6% and 67.3%, respectively. Significantly, patients in the study group demonstrated MRD negativity to a much higher degree (60.6% to 15.6%) than those who received standard therapy.

Equecabtagene autoleucel (Fucaso^®^; CT103A), another BCMA-targeted CAR T-cell therapy, was designated an RMAT (Regenerative Medicine Advanced Therapy) agent by the FDA in 2023, the same year in which it was approved for RRMM by China’s National Medical Products Administration (NMPA). The application for approval was supported by data from the phase I/II FUMANBA-1 trial (NCT05066646; CTR20192510), which found that the agent elicited an ORR of 96% in 101 evaluable subjects, including complete responses or better in 74.3% and a 12-month PFS rate of 78.8% [172].

Other autologous anti-BCMA CAR T-cell products currently under development include (see Table 8):

Zevorcabtagene autoleucel (zevo-cel; CT053): received an RMAT designation for RRMM treatment from the FDA in 2019. One study (NCT03975907) of patients with at least three prior therapies reported an ORR of 100% in 14 subjects [173].CC-98633 (BMS-986354): manufactured using the NEX-T process, which combines shortened manufacturing time with improved potency and phenotypic attributes to enhance the depth and durability of response [174]. This product is similar in construction to orvacabtagene autoleucel (orva-cel), a once-promising product that is no longer being pursued by its sponsor.PHE885: a construct manufactured using the proprietary Novartis T-Charge platform, which is said to enable patient access in under two days with enhanced efficacy [175].C-CAR088: derived employing an scFv obtained from a human IgG1 antibody with a high binding affinity (K_D_ = 0.08 nM) for epitome cluster E3 of the BCMA extracellular domain [176].CART-ddBCMA: a product of Arcelix that utilizes a non-scFv synthetic BCMA-binding domain, known as a D-Domain [177].NXC-201 (HBI0101): similar in structure to cilta-cel and ide-cel in that it incorporates the same 4-1BB and CD3ζ signaling entities [178,179], this BCMA-targeted product is currently under investigation for both RRMM and AL (amyloid light chain) amyloidosis (NCT04720313).

**Table 8 ijms-24-15674-t008:** Selected trials of BCMA-directed CAR T-cell products in multiple myeloma (MM).

Trial ID (References)	Drugs	Phase	Enrollment (N)	Trial Title
NCT03361748 [167]	Ide	II	149	Multicenter Study to Determine the Efficacy and Safety of bb2121 in Subjects with RRMM (KarMMa)
NCT03601078 [180,181]	Ide +/− Len	II	235	Multi-cohort, Open-label, Multicenter Study to Evaluate the Efficacy and Safety of bb2121 in Subjects with RRMM and in Subjects with Clinical High-Risk MM (KarMMa-2)
NCT03651128 [168]	Ide + SOC MM agents	III	381	Multicenter, Randomized, Open-label Study to Compare the Efficacy and Safety of bb2121 Versus Standard Regimens in Subjects with RRMM (KarMMa-3)
NCT04855136	Ide + CC-220 vs. Ide + BMS-986405	I/II	312	Exploratory Trial to Determine Recommended Phase 2 Dose (RP2D), Safety and Preliminary Efficacy of bb2121 (Ide-cel) Combinations in Subjects with RRMM (KarMMa-7)
NCT03548207 [182,183,184]	Cil	I/II	126	Open-Label Study of JNJ-68284528, A Chimeric Antigen Receptor T-Cell (CAR-T) Therapy Directed Against BCMA in Subjects with RRMM (CARTITUDE-1)
NCT04133636 [185,186,187]	Cil + Len + Dara + Dex + Bort	II	169	Multicohort Open-Label Study of JNJ-68284528, a Chimeric Antigen Receptor T Cell (CAR-T) Therapy Directed Against BCMA in Subjects with MM (CARTITUDE-2)
NCT04181827 [171]	Pom + Bort + Dex + Dara +/− Cil	III	419	Randomized Study Comparing JNJ-68284528, a Chimeric Antigen Receptor T Cell (CAR-T) Therapy Directed Against BCMA, Versus Pomalidomide, Bortezomib and Dexamethasone (PVd) or Daratumumab, Pomalidomide and Dexamethasone (DPd) in Subjects with Relapsed and Lenalidomide-Refractory MM (CARTITUDE-4)
NCT04923893 [188]	Bort + Dex + Len + Cil + Ctx + Flu	III	650	Randomized Study Comparing Bortezomib, Lenalidomide and Dexamethasone (VRd) Followed by Ciltacabtagene Autoleucel, a Chimeric Antigen Receptor T Cell (CAR-T) Therapy Directed Against BCMA Versus Bortezomib, Lenalidomide, and Dexamethasone (VRd) Followed by Lenalidomide and Dexamethasone (Rd) Therapy in Participants with NDMM for Whom Hematopoietic Stem Cell Transplant is Not Planned as Initial Therapy (CARTITUDE-5)
NCT05257083 [189]	Dara + Bort + Len + Dex +/− (Cil + Ctx + Flu)	III	750	Randomized Study Comparing Daratumumab, Bortezomib, Lenalidomide and Dexamethasone (DVRd) Followed by Ciltacabtagene Autoleucel Versus Daratumumab, Bortezomib, Lenalidomide and Dexamethasone (DVRd) Followed by ASCT in Participants with NDMM Who Are Transplant Eligible (CARTITUDE-6)
NCT03758417 [190]	Cil	II	130	Open-Label Study of LCAR-B38M CAR-T Cells, a Chimeric Antigen Receptor T-cell (CAR-T) Therapy Directed Against BCMA in Chinese Subjects with RRMM (CARTIFAN-1)
NCT03090659 [191]	Cil	I/II	100	Study of Legend Biotech BCMA-chimeric Antigen Receptor Technology in Treating RRMM Patients (LEGEND-2)
NCT03975907 [173]	Zev	I/II	114	Open Label Trial to Evaluate the Safety and Efficacy of Fully Human Anti-BCMA Chimeric Antibody Receptor Autologous T Cell (CAR T) in Patients with RRMM (LUMMICAR STUDY 1)
NCT03915184 [192,193]	Zev	I/II	105	Open Label, Multi-center Trial to Evaluate the Safety and Efficacy of Autologous CAR BCMA T Cells (CT053) in Patients with RRMM (LUMMICAR STUDY 2)
NCT05066646 [172]	CT103A	I/II	132	Study on Fully Human BCMA Chimeric Antigen Receptor Autologous T Cell Injection (CT103A) in the Treatment of Patients with RRMM (FUMANBA-1)
NCT04318327 [194]	PHE885	I	96	Open Label, Study of B-cell Maturation Antigen (BCMA)-Directed CAR-T Cells in Adult Patients with MM
NCT05172596	PHE885	II	136	Study of PHE885, B-cell Maturation Antigen (BCMA)- Directed CAR-T Cells in Adult Participants with RRMM.
NCT05521802	C-CAR088	I/II	92	Study of CBM.BCMA Chimeric Antigen Receptor T Cell Product (C-CAR088) for Treating Patients with RRMM
NCT04295018 [176]	C-CAR088	I	10	A Study Evaluating Safety and Efficacy of C-CAR088 Treatment in Subjects with RRMM
NCT05396885 [195]	CART-ddBCMA	II	110	Study of CART-ddBCMA for the Treatment of Patients with RRMM (IMMagine-1)
NCT04394650 [196]	CC-98633/BMS-986354	I	150	Multi Center, Open Label Study of CC-98633, BCMA Targeted NEX-T Chimeric Antigen Receptor (CAR) T Cells, in Subjects with RRMM
NCT04720313 [178]	NXC-201	I	160	Dose Escalation and Safety Study of NXC-201 (Formerly HBI0101) CART in BCMA-Expressing MM Patients and AL Amyloidosis

AL = amyloid light chain; Bort = bortezomib; Cil = ciltacabtagene autoleucel; Ctx = cyclophosphamide; Dara = daratumumab; Dex = dexamethasone; Flu = fludarabine; Ide = idecabtagene vicleucel; Len = lenalidomide; MM = multiple myeloma; NDMM = newly diagnosed multiple myeloma; Pom = pomalidomide; RRMM = relapsed and/or refractory multiple myeloma; SOC = standard of care; Zev = zevorcabtagene autoleucel.

While impressive results in terms of depth and duration of response have been obtained in many cases with BCMA-directed T-cell therapies, relapse is a frequent occurrence. A potential limitation of BCMA-targeted immunotherapy is the heterogeneous nature of the antigen’s expression on myeloma cells, as well as its reduction or complete loss during drug exposure. GC012F, which targets both BCMA and CD19 is an attempt to circumvent this issue. This dual-targeting autologous product, which is currently the subject of six different RRMM trials, can be manufactured in 36 h or less using the FasTCAR platform, according to its innovator Gracell [197,198]. In one such trial (NCT04236011/NCT04182581), MRD negativity was reported to occur in 21/28 RRMM patients after 28 days of a single infusion [199]. A study of the drug included as a component of standard therapy in 13 transplant-eligible high-risk NDMM patients (NCT04935580) attained 100% for both ORR and MRD negativity after a median of 5.3 months [198]. 

As seen with the BiTE therapy, GPRC5D has also begun to play a major role in the development of new CAR T-cell-targeted therapeutics for RRMM [200]. One of the most promising of the GPRC5D-targeted agents is MCARH109, the subject of an ongoing phase I trial (NCT04555551). An ORR of 71% (10/17) was reported for the entire cohort in that study. While the drug was well tolerated generally, a single patient did experience both grade 4 CRS and ICANS [201]. Another entry in the anti-GPRC5D CAR T-cell arena is BMS-986393 (CC-95266), currently the focus of a phase I dose-escalation study (NCT04674813). An initial report of evaluable patients found an ORR of 86% (12/14) after one-month of therapy. Significantly, eight of the patients displayed high-risk cytogenetics—del[17p], t[4;14], and/or t[14;16] [202]. NCT05016778 is yet another phase I study evaluating a GPRC5D-directed CAR T-cell preparation, named OriCAR-017. Data from that trial, titled POLARIS, reported a 100% response rate on the first 10 patients evaluated [203]. Finally, CAR T-cells targeting SLAMF7 (NCT03958656 and NCT03710421) and CD38 (NCT03464916) have received some attention based on preclinical studies but have resulted in no published patient data.

Alternatives to T-cells have also been under consideration in efforts to expand available options in the CAR arena for RRMM, the primary candidates being autologous anti-BCMA engineered NK cells [204]. An examination of the NCI clinical trials data base uncovered two such studies (NCT03940833 and NCT05008536), although neither has yielded published data thus far.

Even though autologously administered CAR T-cells have thus far produced highly promising results, their use is not without drawbacks, such as the short durations of response they elicit, the extended time and complexity involved in their manufacture, as well as the high risk of CRS and other dose-limiting adverse events. Consequently, some CAR T-cell originators have pioneered programs to develop “off-the-shelf” allogeneic anti-BCMA products that employ T-cells from healthy donors. Although efficacy and toxicity data have been sparse so far, the potential feasibility of this approach to RRMM therapy continues to undergo intense scrutiny with the current pipeline of agents in this category being headed by three major players: ALLO-715 from Allogene Therapeutics (San Francisco, CA, USA), Precision Bioscience’s PBCAR269A (Durham, NC, USA), and Caribou Bioscience’s CB-011 (Berkeley, CA, USA).

ALLO-715 is a product of the proprietary Transcription Activator-like Effector Nuclease (TALEN^®^) system, a site-specific BCMA gene-editing technique that is hypothesized to limit T-cell receptor-mediated immune reactions, including rapid rejection and graft-versus-host disease (GvHD) [205]. Data from the phase I UNIVERSAL trial (NCT04093596) recently reported an ORR of 55.8% (24/43) in patients who received ALLO-715 following lymphodepletion that included the anti-CD52 antibody ALLO- 647 in addition to cyclophosphamide/fludarabine [206]. It is noteworthy that the design strategy of ALLO-715 also incorporates an off-switch, a CD20-based mimotope capable of inactivation by rituximab. The Arcus gene-editing platform, which is based on the proprietary homing endonuclease I-Crel scaffold [207], serves as the foundation for the production of PBCAR269A. This allogeneic product currently is under study in a phase I/II trial (NCT04171843) that also includes the gamma-secretase inhibitor nirogacestat. As yet, there has been no published patient data concerning that study. The CRISPR-edited production of CB-011 was recently outlined by Berdeja et al. [208], who also described the methodology used in the ongoing clinical evaluation of this agent (NCT05722418; CaMMouflage). Two additional BCMA-directed allogeneic CAR candidates now under investigation for RRMM but with no reported results are Allogene’s ALLO-605 (NCT05000450) [209] and Poseidon’s P-BCMA-ALLO1 (NCT04960579) [210]. Significantly, the latter product is marked by inclusion in its design of a truncated caspase 9 domain as a safety switch that can activate apoptosis upon administration of the caspase 9 dimerizing agent rimiducid if patient safety is threatened [211]. It should be noted that the construction of P-BCMA-ALLO1 was informed by the promising initial results [212] obtained with the company’s autologous entry into the field, P-BCMA-101, which has since been superseded in development by the allogeneic product [213]. Finally, the potential of a few other allogeneic CAR products that are in early-stage development for RRMM has been described; these include the SLAMF7/CS1-directed UCARTCS1 (NCT04142619; MELANI-01) [214], a dual targeting BCMA/GPRC5D NK cell preparation [215], and FT538, a multiplex engineered NK product derived from induced pluripotent stem cells (NCT04614636) [216].

Table 9 summarizes both efficacy and toxicity data as found in ten different meta-analytic studies of BCMA-targeted CAR T-cell trials and indicates the high degree of effectiveness and safety of the various autologous products described above. In the years ahead, similar studies comparing the autologous products with the now-emerging allogeneic preparations will help illuminate the path forward concerning the role of ACT in the treatment of RRMM.

## 7. Cytokine Release Syndrome and Neurotoxicity

The potential to produce severe life-threatening toxicity, often divided into two general types, is a major challenge confronting the use of immunotherapeutic measures in the treatment of MM and other hematologic cancers. The first type results from T-cell recognition and activation against the targeted malignant cells followed by uncontrolled release of high levels of cytokines (on-target, on-tumor toxicity). In the second case, the released cytokines bind to the target antigen located on normal cells (on-target, off-tumor toxicity). A low risk of on-target, off-tumor toxicity attends the use of the antimyeloma BCMA-targeted agents described in this review since the distribution of the target antigen is confined virtually to plasma cells. On the other hand, on-tumor, on-target toxicities can be very serious since they often are not easily reversed and may require intensive multidisciplinary care. The discussion that follows concerns the on-target, on-tumor types of adverse reactions.

The most important adverse effects that accompany CAR T-cell therapy are the cytokine release syndrome (cytokine storm; CRS) and neurotoxicity, both of which are observed in varying degrees of severity in a percentage of participants in every trial involving CAR T-cell products. The neurotoxic reactions are often referenced in the literature by either of two synonymous terms: CAR T-cell-related encephalopathy syndrome (CRES) or immune effector cell-associated neurotoxicity syndrome (ICANS).

Generally occurring within the first two weeks of therapy and often resembling a severe inflammatory reaction, the symptoms of CRS are attributed to a marked increased expression and release of certain cytokines, including IL-6, IL-2R, IL-10, IFN-γ, and TNF-α [227]. The gradation of CRS symptoms is based on a 1–4 system with grade 4 having severe life-threatening consequences [228]. Moreover, the CRS has been implicated as a major contributor to the acute respiratory distress syndrome and multiorgan failure accompanying COVID-19 infections [229]. The IL-6 blocker tocilizumab (Actemra^®^), an IL-6 blocker often given with corticosteroids, is the treatment of choice for cases of CRS associated with CAR T-cell therapy. FDA approval of tocilizumab for this indication was granted in 2017 [230]. However, the efficacy of combining corticosteroids with tocilizumab in the management of CRS has not been investigated through randomized controlled studies. A recent metanalysis of 53 studies involving a total of 2092 RRMM patients comparing BCMA-targeting bispecific antibody with CAR T-cell therapies showed that the latter are associated with higher rates and longer durations of grade ≥ 3 CRS and a more frequent use of tocilizumab/corticosteroid [231].

In order to help mitigate toxicity arising from CAR T-cell therapy some products have incorporated a safety switch to enable activity to be curtailed by pharmacological means [232,233]. One early example is the inclusion within the construct of a CD20 receptor that can be turned “off” by the administration of the CD20 blocker rituximab [234]. The incorporation of a nonfunctional truncated epidermal growth factor receptor (tEGFR) inhibitable by cetuximab represents another illustration [235]. The dimerization of caspase-9 by rimiducid to activate apoptosis is yet another example [236]. The various types of molecular safety switches employed in CAR T-cell design, including the strengths and weaknesses of each, have been discussed in a recent review [237].

## 8. Checkpoint Inhibitors

Immune checkpoint blockade has emerged as a major approach to anticancer drug development over the past decade. The strategy has as its foundation the interaction of specific cell surface biomarkers and their cognate ligands, which enable malignant cells to evade immune surveillance and elimination. The development of mAbs to block these biomarker/ligand interactions remains a major focus of cancer immunotherapy with the bulk of attention focused on two T-cell surface proteins: CTLA-4 (cytotoxic T-lymphocyte–associated protein-4) and the programmed death (PD) receptor and their corresponding ligands, referred to as B7-1/B7-2 and PD-L1, respectively, on targeted tumor cells. These receptor/ligand pairs have been employed successfully as targets for a number of different cancers including non-small cell lung cancer, renal cell carcinoma, melanomas, and Hodgkin’s lymphoma [238]; however, their utility as targets in leukemias and MM has yet to be conclusively demonstrated [239]. A myeloma-based trial (NCT02681302) of the anti-CTLA-4 mAb ipilimumab combined with another checkpoint inhibitor, nivolumab, is currently ongoing. An initial report from this study concluded that the combined checkpoint inhibitor therapy was safe and had the potential to increase the “depth of response in patients with high-risk disease” [240]. In a major development, two phase III studies (NCT02576977 KEYNOTE 183 and NCT02579863 KEYNOTE 185) [241,242] of the PD-1 blocker pembrolizumab combined with immunomodulators in MM had to be terminated when decreased OS was found in both trials [243]. Although the mechanism underlying this safety concern remains unknown, several trials of pembrolizumab either as sole therapy or in combination with established antimyeloma agents remain in effect (see Table 10).

Also under clinical study for MM are the PD-1 blockers nivolumab and cemiplimab and the PD-L1 inhibitor atezolizumab. A trial (NCT01592370) that incorporated a total of 81 hematologic cancer patients found that the RRMM (N = 27) cohort received negligible clinical benefit. (PFS = 10 weeks; ORR = 4%) [244] from nivolumab monotherapy. Another study (NCT02681302) of seven high-risk transplant-naïve MM patients receiving combined checkpoint inhibitor therapy (ipilimumab + nivolumab) reported OS and PFS rates of 40% and 57.1% after 18 months [240]. In another instance, data acquired before the trial (NCT03312530) was terminated early demonstrated that atezolizumab provided some minor benefit when combined with the MEK inhibitor cobimetinib and the BCl2 blocker venetoclax in RRMM patients [245]. Likewise, only marginal benefits were realized in a trial in which isatuximab and cemiplimab were employed (NCT03194867) as treatment in 106 RRMM subjects [246].

Like T-cells, macrophages, members of the innate immune system, are also known to express a checkpoint receptor known as signal regulatory protein a (SIRPα or CD172α), which attaches to its ligand CD47, a transmembrane protein overexpressed by cancer cells, including myeloma cells. When SIRPα and CD47 interact, a signaling cascade is initiated leading to the suppression of macrophage phagocytic activity, sometimes known as the “don’t eat me” signal [247]. Recently, efforts have emerged to target this immune checkpoint in the search for novel approaches to treat MM [248]. One of these, the CD47 blocker magrolimab, is the subject of a current trial (NCT04892446) in combination with various established antimyeloma agents [249]. Clinical investigations have also begun on two additional anti-CD47 mAbs: AO-176 (NCT04445701) [250] and TTI-622 (NCT05139225) (see Table 10). However, no data have yet appeared to substantiate the clinical validity of the CD47-targeted approach to MM.

**Table 10 ijms-24-15674-t010:** Selected current trials of checkpoint inhibitors in MM.

PD-1 Inhibitors				
Trial ID (References)	Drugs	Phase	Enrollment (N)	Trial Title
NCT05204160	Pembro	II	30	Study of Pembrolizumab as Salvage Therapy Among MM Patients Progressing on CAR-T Cell Therapy
NCT05191472 [251]	Pembro	II	25	Study of Pembrolizumab in MM Patients Relapsing After or Refractory to Anti-BCMA CAR-T Therapies
NCT02636010	Pembro	II	20	Multicenter, Open Label, Clinical Trial of Pembrolizumab as Consolidation Therapy in MM Patients with Residual Disease After Treatment
NCT03267888	Pembro	I	26	Pilot Study of Pembrolizumab and Single-Fraction, Low-Dose, Radiation Therapy in Patients with RRMM
NCT02331368	Pembro	II	32	Multi-center Study of Anti-Programmed-Death-1 During Lymphopenic State After High-Dose Chemotherapy and ASCT for MM
NCT05514990	Pembro + Bort + Dex +/− Pelareorep	I/II	42	Study of Standard Doses of Bortezomib and Pembrolizumab ± Reovirus (Pelareorep) Combination Therapy in Patients with RRMM (AMBUSH Study)
NCT02906332 [252]	Pembro + Len + Dex	II	16	Trial of Pembrolizumab + Lenalidomide + Dexamethasone as Post-ASCT Consolidation in Patients with High-risk MM
NCT03848845 [62]	Pembro + BelMaf	I/II	41	Single Arm Open-Label Study to Explore Safety and Clinical Activity of GSK2857916 Administered in Combination with Pembrolizumab in Subjects with RRMM (DREAMM 4)
NCT02880228	Pembro + Len + Dex	II	11	Trial of Pembrolizumab, Lenalidomide, and Dexamethasone for Initial Therapy of NDMM Eligible for ASCT
NCT03506360	Pembro + Ixaz + Dex	II	13	Trial of Pembrolizumab, Ixazomib, and Dexamethasone for RRMM
NCT03194867 [246]	Isa +/− Cem	I/II	106	Study to Evaluate Safety, Pharmacokinetics and Efficacy of Isatuximab in Combination with Cemiplimab in Patients with RRMM
NCT03184194	Niv + Dara +/− Ctx	II	62	Study of Nivolumab Combined with Daratumumab with or without Low-dose Cyclophosphamide in RRMM
NCT03605719	Niv + Carf + Dex +/− Pelareorep	I	23	Dexamethasone, Carfilzomib, and Nivolumab with Pelareorep for RRMM
NCT04119336	Niv + Ixaz + Ctx + Dex	II	50	Study of Nivolumab in Combination with Ixazomib, Cyclophosphamide, and Dexamethasone in RRMM
NCT02726581	Pom + Dex +/− Niv	III	170	Open-Label, Randomized Trial of Combinations of Nivolumab, Pomalidomide and Dexamethasone in RRMM
NCT02681302 [240] ^a^	Niv + Ipil	I/II	35	Study of Combined Check Point Inhibition After ASCT in Patients at High Risk for Post-transplant Recurrence
NCT02612779	(Niv + Elo) vs. (Niv + Elo + Pom + Dex)	II	74	Study of Elotuzumab in Combination with Pomalidomide and Low Dose Dexamethasone and Elotuzumab in Combination with Nivolumab in Patients with RRMM to Prior Treatment with Lenalidomide.
NCT03292263	Mel + Niv	I/II	30	ASCT with Nivolumab in Patients with MM
NCT01592370 [244]	Niv vs. (Niv + Ipil) vs. (Niv + Liri) vs. (Niv + Dara + Pom) + Dex) vs. (Niv + Dara)	I/Ii	316	An Investigational Immuno-Therapy Study to Determine the Safety and Effectiveness of Nivolumab and Daratumumab in Patients with MM
PD-L1 Inhibitors				
NCT03312530 [245]	(Cobi + Atez) vs. (Cobi + Ven +/− Atez)	I/II	49	Study of Cobimetinib Administered as Single Agent and in Combination with Venetoclax, with or without Atezolizumab, in Patients with RRMM
NCT02431208 [253]	(Atez + Len) vs. (Atez + Dara) vs. (Atez + Dara + Len) vs. (Atez + Dara + Pom) vs. (Dara + Pom + Dex)	I	85	Study of the Safety and Pharmacokinetics of Atezolizumab Alone or in Combination with an Immunomodulatory Drug and/or Daratumumab in Patients with RRMM and ASCT
CD47 Inhibitors				
NCT04892446 [249,254]	(Mag + Dara) vs. (Mag + Pom + Dex) vs. (Mag + Carf + Dex) vs. (Mag + Bort + Dex)	II	153	Multi-Arm Study of Magrolimab Combinations in Patients with RRMM
NCT04445701 [250]	AO-176 vs. (AO-176 + Dex +/− Bort)	I/II	157	Dose Escalation Safety and Tolerability Study of AO-176 as Monotherapy and in Combination with Bortezomib and Dexamethasone in Adults with RRMM
NCT05139225	TTI-622 + Dara-Hyal	I	32	Study Of The Combination Of CD47 Blockade with SIRP-Alpha FC Fusion Proteins (TTI-622) And Daratumumab Hyaluronidase-fihj For Patients with RRMM

ASCT = autologous stem cell transplantation; Atez = atezolizumab; BelMaf = belantamab mafodotin; Bort = bortezomib; Cem = cemiplimab; Carf = carfilzomib; Cobi = cobimetinib; Ctx = cyclophosphamide; Dara = daratumumab; Dex = dexamethasone; Elo = elotuzumab; Hyal = hyaluronidase; Ipil = ipilimumab; Isa = isatuximab; Ixaz = ixazomib; Len = lenalidomide; Mag = magrolimab; Mel = melphalan; MM = multiple myeloma; NDMM = newly diagnosed multiple myeloma; Niv = nivolumab; Pembro = pembrolizumab; Pom = pomalidomide; RRMM = relapsed and/or refractory multiple myeloma; Ven = venetoclax. ^a^ The study includes patients with lymphomas and high-risk recurrent MM.

## 9. Summary and Conclusions

The therapeutic measures used to treat MM have undergone a fundamental shift since the beginning of the current century with the introduction of several entirely new classes of agents, such as proteasome inhibitors, immunomodulators, nuclear export blockers, and apoptosis inducers. These advances are reflected in the dramatic improvements in the 5-year survival rates for the disease, which had risen gradually from 25% in the 1975–77 period to 32% for 1995–97 but climbed sharply to 58% during 2012–2018 as the impact of these newer modalities began to come into play [2]. The benefits of these therapeutic advances are further evidenced by such measures as accumulated survival and response data and studies demonstrating an improved quality of life [255,256]. The characterization of MM itself has changed over the years, shifting from the traditional CRAB model toward a paradigm that includes molecular and immunologically based factors that may be evidence of the disease before signs and symptoms emerge.

These criteria, which include bone marrow studies and imaging technologies, as well as free light chain, immunoglobulin, and cytogenetic analyses, have been woven into the fabric of contemporary diagnostic procedures, projection of disease course, and evaluation of the expanding number of available intervention strategies tailored to the individual patient [6].

The 2015 FDA approval of daratumumab heralded the advent of the era of immunologic-based approaches to MM as this monoclonal antibody quickly rose to a position of prominence as a core component in the treatment of MM, whether newly diagnosed or at the relapsed/refractory stage. Since that seminal event, the immunologic tools available in the clinic to treat MM have greatly expanded to include additional mAbs, such as isatuximab, bispecifics, and CAR-T products. However, this sense of optimism is balanced by the reality that MM still is considered incurable regardless of the therapeutic measures employed. Initial therapeutic benefits are all too frequently followed by refractoriness to treatment and a resulting relapse.

The identification of suitable tumor-driving biomarkers for the disease that can serve as potential therapeutic targets is fundamental to immunotherapeutic approaches to MM. Currently, the surface molecule in the vanguard of this pursuit is BCMA, which accounts for most of the immunologically based clinical trials currently in progress. While several studies of BCMA-targeted CAR T-cell therapies have been shown to provide ORRs of at least 80%, the response durations tend to be short. Moreover, BCMA expression by myeloma cells may not be as homogeneous as previously believed. For example, in one study a substantial proportion (33/85) of MM patients was found to be BCMA-negative [257]. In addition, downregulated BCMA expression by tumor cells has been reported during CAR T-cell therapy, as well as instances in which a significant fraction of initially responding patients experienced relapse despite a continued expression of BCMA [258,259,260,261]. Also relevant to the discussion is a genome-wide gene-editing CRISPR study that failed to identify BCMA as being included among 90 different genes essential for MM [262]. As a consequence, there is increasing interest in combinatorial immunologically based strategies that employ BCMA with other validated myeloma targets, such as CD38, SLAMF7, and CD19 [263] or with emerging viable targets like GPRC5D [215,264]. Indeed, GPRC5D has a special appeal as a target in immunotherapy constructs since its expression is linked with poor prognosis in MM [132] and, unlike BCMA, does not shed from the cell surface. In addition, GPRC5D expression is independent of BCMA, suggesting that it may serve as an alternative target in cases of anti-BCMA therapy relapse [265]. The recent approval of the GPRC5D x CD3 bispecific talquetamab for MM is a promising sign in this regard that portends the possibility of potentially fruitful avenues for the discovery of innovative approaches to treat RRMM based on this biomarker.

The development of new and ground-breaking therapies to treat RRMM will also depend on gaining a much better insight into the mechanisms that drive the resistance to treatment, especially the role played by the bone marrow microenvironment and the factors that enable malignant plasma cells to escape immune surveillance. Advances in understanding these basic mechanisms will translate into improved outcomes, including the ability to prevent bone loss [266].

In addition to ORR, CR, OS, PFS, and other assessments, the ability to detect minimal residual disease (MRD), i.e., a complete absence of any detectable clonal plasma cells, has emerged as a key factor in directing antimyeloma therapies enabled by technologic advances. The attainment of MRD negativity is increasingly seen as a reliable marker and strategic objective in both clinical trials and in the monitoring of individual patients receiving standard care [267].

Finally, although ASCT has been a standard of care for transplant-eligible MM patients since its introduction nearly forty years ago, its hematopoietic source counterpart, namely, allogeneic stem cell transplantation (allo-SCT), has received comparatively little attention until rather recently. Allo-SCT, which has been described as having the potential to yield curative outcomes in the disease, is based on the graft vs. myeloma effect first described in 1996 [268]. While at present this modality is generally reserved for young patients having high-risk relapsed myeloma, it remains controversial due to its overall mixed efficacy and lack of clear treatment guidelines [269]. On the other hand, this approach to MM therapy bears watching for future developments and possibly wider applications [270].

In conclusion, it is most encouraging to note that the five-year survival rate for MM is now approaching 60%, based on data that only begin to capture outcomes from the present era in which monoclonal antibodies, such as daratumumab, have assumed a much greater therapeutic role in the disease. The vast expansion over the past two decades in understanding the mechanisms that underlie MM initiation, proliferation, survival, and metastasis along with the identification of druggable biomarkers that have led to increasingly efficacious therapies has elevated the disease to its current status as a manageable, albeit still incurable, chronic condition. Nevertheless, the field continues to be burdened with the challenges of relapse, as patients become refractory to treatment and eventually succumb to the disease. Moving forward, confronting these challenges will need to continue with sustained and vigorous resolve in order for the goal of a true cure to become a future reality.

## Figures and Tables

**Table 1 ijms-24-15674-t001:** Selected trials of belantamab mafodotin in multiple myeloma (MM).

Trial ID (References)	Drugs	Phase	Enrollment (N)	Trial Title
NCT02064387 [57]	BelMaf	I	79	Open-Label, Dose Escalation Study To Investigate The Safety, Pharmacokinetics, Pharmacodynamics, Immunogenicity And Clinical Activity Of The Antibody Drug Conjugate GSK2857916 In Subjects with RRMM And Other Advanced Hematologic Malignancies Expressing BCMA (DREAMM -1)
NCT03525678 [55,58,59,60]	BelMaf	II	221	Open Label, Randomized, Two-Arm Study To Investigate The Efficacy And Safety Of Two Doses Of The Antibody Drug Conjugate GSK2857916 In Participants with MM Who Had 3 Or More Prior Lines Of Treatment, Are Refractory To A Proteasome Inhibitor And An Immunomodulatory Agent And Have Failed An Anti-CD38 Antibody (DREAMM-2)
NCT04162210 [61]	BelMaf vs. (Pom + Dex)	III	180	Open-Label, Randomized Study To Evaluate The Efficacy And Safety Of Single Agent Belantamab Mafodotin Compared To Pomalidomide Plus Low Dose Dexamethasone In Participants with RRMM (DREAMM-3)
NCT03848845 [62]	BelMaf + Pembro	I/II	41	Single Arm Open-Label Study To Explore Safety And Clinical Activity Of GSK2857916 Administered In Combination with Pembrolizumab In Subjects with RRMM (DREAMM-4)
NCT04126200 [52,63,64]	BelMaf + Various	I/II	464	Randomized, Open-label Platform Study Utilizing a Master Protocol to Study Belantamab Mafodotin (GSK2857916) as Monotherapy and in Combination with Anti-Cancer Treatments in Participants with RRMM—DREAMM-5
NCT03544281 [65,66,67]	BelMaf + Dex + (Len or Bort)	I/II	152	Open-Label, Dose Escalation And Expansion Study To Evaluate Safety, Tolerability, And Clinical Activity Of GSK2857916 Administered In Combination with Lenalidomide Plus Dexamethasone, Or Bortezomib Plus Dexamethasone In Participants with RRMM (DREAMM-6)
NCT04246047 [68,69]	Bort + Dex + (BelMaf or Dara)	III	478	Multicenter, Open-Label, Randomized Study To Evaluate The Efficacy And Safety Of The Combination Of Belantamab Mafodotin, Bortezomib, And Dexamethasone Compared with The Combination Of Daratumumab, Bortezomib, And Dexamethasone In Participants with RRMM (DREAMM-7)
NCT04484623 [70,71]	Pom + Dex + (BelMaf or Bort)	III	450	Multicenter, Open-Label, Randomized Study To Evaluate The Efficacy And Safety Of Belantamab Mafodotin In Combination with Pomalidomide And Dexamethasone Versus Pomalidomide Plus Bortezomib And Dexamethasone In Participants with RRMM (DREAMM-8)
NCT04091126 [72,73,74]	Bort + Len + Dex (± BelMaf)	I	144	Randomized, Dose And Schedule Evaluation Study To Investigate The Safety, Pharmacokinetics, Pharmacodynamics And Clinical Activity Of Belantamab Mafodotin Administered In Combination with Standard Of Care In Participants with NDMM (DREAMM-9)
NCT05064358 [75]	BelMaf	II	180	Randomized, Parallel, Open-label Study to Investigate the Safety, Efficacy, and Pharmacokinetics of Various Dosing Regimens of Single-agent Belantamab Mafodotin (GSK2857916) in Participants with RRMM (DREAMM-14)

BelMaf = belantamab mafodotin; Bort = bortezomib; Dara, = daratumumab; Dex = dexamethasone; Len = lenalidomide; MM = multiple myeloma; NDMM = newly diagnosed MM; Pembro = pembrolizumab; Pom = pomalidomide; RRMM = relapsed and/or refractory MM.

**Table 2 ijms-24-15674-t002:** Non-BCMA-directed ADCs in Active Clinical Trials for RRMM.

Drug	Target	Trial ID (References)	Payload
STI-6129	CD38	NCT05308225 [85]	Duostatin-5.2 (MMAF derivative; tubulin inhibitor)
Modakafusp alfa (TAK-573)	CD38	NCT03215030 [86]	Interferon alfa 2b (attenuated form)
MT-169 (TAK-169)	CD38	NCT04017130 [87]	Ribosome-inactivating Shiga-like toxin-A subunit
FOR46	CD46	NCT03650491 [88]	MMAE
Indatuximab ravtansine (BT-062)	CD138	NCT01638936 [89]	DM4 (antitubulin maytansinoid)
LM-305	GPRC5D	NCT05647512 [90]	NA
STRO-001	CD74	NCT03424603 [91,92]	Maytansinoid (chemistry not disclosed)

MMAE = monomethylauristatin E; MMAF = monomethylauristatin F; NA = information not available.

**Table 3 ijms-24-15674-t003:** Selected trials of teclistamab in multiple myeloma (MM).

Trial ID (References)	Drugs	Phase	Enrollment (N)	Trial Title
NCT04557098 [95,96,97,98]	Tec	I/II	244	First-in-Human, Open-Label, Dose Escalation Study of Teclistamab in Subjects with RRMM (MajesTEC-1)
NCT03145181 [99]	Tec iv vs. Tec sc	I	282	First-in-Human, Open-Label, Dose Escalation Study of Teclistamab, a Humanized BCMA x CD3 Bispecific Antibody in Subjects with RRMM (MajesTEC-1)
NCT04722146 [100,101]	Tec in various combinations with antimyeloma agents	Ib	140	A Multi-arm Study of Teclistamab with Other Anticancer Therapies in Participants with MM (MajesTEC-2)
NCT05083169 [102]	(Dara + Tec) vs. (Dara + Pom + Bort + Dex)	III	560	Randomized Study Comparing Teclistamab in Combination with Daratumumab SC (Tec-Dara) Versus Daratumumab SC, Pomalidomide, and Dexamethasone (DPd) or Daratumumab SC, Bortezomib, and Dexamethasone (DVd) in Participants with RRMM (MajesTEC-3)
NCT05243797 [103]	(Tec +/− Len) vs. Len	III	1500	Teclistamab in Combination with Lenalidomide and Teclistamab Alone Versus Lenalidomide Alone in Participants with NDMM as Maintenance Therapy Following Autologous Stem Cell Transplantation—MajesTEC-4
NCT05552222 [104]	Tec + Dara + Len vs. Dara + Len +Dex	III	1060	Randomized Study Comparing Teclistamab in Combination with Daratumumab SC and Lenalidomide (Tec-DR) Versus Daratumumab SC, Lenalidomide, and Dexamethasone (DRd) in Participants with NDMM Who Are Either Ineligible or Not Intended for Autologous Stem Cell Transplant as Initial Therapy (MajesTEC-7)
NCT05572515	Tec vs. (Pom + Bort + Dex or Carf + Dex)	III	590	Randomized Study Comparing Teclistamab Monotherapy Versus Pomalidomide, Bortezomib, Dexamethasone (PVd) or Carfilzomib, Dexamethasone (Kd) in Participants with RRMM Who Have Received 1 to 3 Prior Lines of Therapy, Including an Anti-CD38 Monoclonal Antibody and Lenalidomide (MajesTEC-9)
NCT04586426	Tec + Tal + Dara	I/II	164	Dose Escalation Study of the Combination of the Bispecific T Cell Redirection Antibodies Talquetamab and Teclistamab in Participants with RRMM (RedirecTT-1)
NCT04108195 [105,106]	(Dara + Tec +/− Pom) vs. (Dara + Tal +/− Pom)	I	295	Study of Subcutaneous Daratumumab Regimens in Combination with Bispecific T Cell Redirection Antibodies for the Treatment of Subjects with MM (TRIMM-2)
NCT05338775	Tec + Tal + “PD-1 inhibitor”	Ib	152	Study of Bispecific T Cell Redirection Antibodies in Combination with Checkpoint Inhibition for the Treatment of Participants with RRMM (TRIMM-3)

Bort = bortezomib; Carf = carfilzomib; Dara = daratumumab; Dex = dexamethasone; Len = lenalidomide; MM = multiple myeloma; NDMM = newly diagnosed MM; Pom = pomalidomide; RRMM = relapsed and/or refractory MM; Tal = talquetamab; Tec = teclistamab.

**Table 4 ijms-24-15674-t004:** Selected trials of elranatamab in multiple myeloma (MM).

Trial ID (References)	Drugs	Phase	Enrollment (N)	Trial Title
NCT03269136 [109,110]	Elran + (Dex or Len or Pom)	I	101	Open Label Study To Evaluate The Safety, Pharmacokinetic, Pharmacodynamic And Clinical Activity Of Elranatamab (PF-06863135), A B-Cell Maturation Antigen (BCMA) X CD3 Bispecific Antibody, As A Single Agent And In Combination with Immunomodulatory Agents In Patients with RRMM (MagnetisMM-1)
NCT04649359 [108]	Elran	II	187	Open-Label, Multicenter, Non-Randomized Study Of Elranatamab (Pf-06863135) Monotherapy In Participants with MM Who Are Refractory To At Least One Proteasome Inhibitor, One Immunomodulatory Drug And One Anti-CD38 Antibody (MagnetisMM-3)
NCT05090566 [111]	(Elran + Niro) vs. (Elran + Len + Dex)	II	105	Open Label Umbrella Study Of Elranatamab (PF-06863135), A B-Cell Maturation Antigen (BCMA) CD3 Bispecific Antibody, In Combination with Other Anti-Cancer Treatments In Participants with MM (MagnetisMM-4)
NCT05020236 [112,113]	Elran vs. (Elran + Dara) vs. (Dara + Pom + Dex)	III	589	Open-Label, 3-Arm, Multicenter, Randomized Study To Evaluate The Efficacy And Safety Of Elranatamab (PF-06863135) Monotherapy And Elranatamab + Daratumumab Versus Daratumumab + Pomalidomide + Dexamethasone In Participants with RRMM Who Have Received At Least 1 Prior Line Of Therapy Including Lenalidomide And A Proteasome Inhibitor (MagnetisMM-5)
NCT05623020	(Elran + Dara + Len) vs. (Dara + Len + Dex)	III	966	Open-Label, 2-Arm, Multicenter, Randomized Study To Evaluate The Efficacy And Safety Of Elranatamab (PF-06863135) + Daratumumab + Lenalidomide Versus Daratumumab + Lenalidomide + Dexamethasone In Transplant-Ineligible Participants with NDMM (MagnetisMM-6)
NCT05317416 [114]	Elran vs. Len	III	760	Randomized, 2-Arm Study Of Elranatamab (PF-06863135) Versus Lenalidomide In Patients NDMM After Undergoing ASCT (MagnetisMM-7)
NCT05014412	Elran vs. Elran + Dex	I/II	76	Open-Label, Multicenter Study To Evaluate A Dosing Regimen with Two Step-Up Priming Doses And Longer Dosing Intervals Of Elranatamab (PF-06863135) Monotherapy In Participants with RRMM (MagnetisMM-9)

ASCT = autologous stem cell transplantation; Dara = daratumumab; Dex = dexamethasone; Elran = elranatamab; Len = lenalidomide; MM = multiple myeloma; NDMM = newly diagnosed MM; Niro = nirogacestat; Pom = pomalidomide; RRMM = relapsed and/or refractory MM.

**Table 6 ijms-24-15674-t006:** Selected trials of talquetamab in multiple myeloma (MM).

Trial ID (References)	Drugs	Phase	Enrollment (N)	Trial Title
NCT05050097	(Tal + Carf +/− Dara) vs. (Tal + Len +/− Dara) vs. (Tal + Pom)	I	182	Multi-arm Study of Talquetamab with Other Anticancer Therapies in Participants with MM (MonumenTAL-2)
NCT05455320 [134]	(Dara + Pom + Dex +/− Tal) vs. (Dara + Dex + Tal)	III	810	Randomized Study Comparing Talquetamab in Combination with Daratumumab (SC) and Pomalidomide (Tal-DP) or Talquetamab (SC) in Combination with Daratumumab SC (Tal-D) Versus Daratumumab SC, Pomalidomide and Dexamethasone (DPd), in Participants with RRMM Who Have Received at Least 1 Prior Line of Therapy (MonumenTAL-3)
NCT04586426	Tal + Tec +/− Dara	I/II	184	Dose Escalation Study of the Combination of the Bispecific T Cell Redirection Antibodies Talquetamab and Teclistamab in Participants with RRMM
NCT05338775	Tal + Tec + PD-1 inhibitor	I	152	Study of Bispecific T Cell Redirection Antibodies in Combination with Checkpoint Inhibition for the Treatment of Participants with RRMM
NCT04634552/NCT03399799 [135,136,137,138,139]	Tal	I/II	320	First-in-Human, Open-Label, Dose Escalation Study of Talquetamab, a Humanized GPRC5D x CD3 Bispecific Antibody, in Subjects with RRMM (MonumenTAL-1)
NCT04108195 [140,141]	(Dara + Tal +/− Pom) vs. (Dara + Tec +/− Pom)	I	294	Study of Subcutaneous Daratumumab Regimens in Combination with Bispecific T Cell Redirection Antibodies for the Treatment of Subjects with MM (TRIMM-2)

Carf = carfilzomib; Dara = daratumumab; Dex = dexamethasone; Len = lenalidomide; Pom = pomalidomide; RRMM = relapsed and/or refractory multiple myeloma; MM = multiple myeloma; Tal = talquetamab; Tec = teclistamab.

**Table 7 ijms-24-15674-t007:** Selective active trials of cevostamab in multiple myeloma (MM).

Trial ID (References)	Drugs	Phase	Enrollment (N)	Trial Title
NCT05801939	Cev	II	30	Study of Cevostamab Consolidation Following BCMA CAR T Cell Therapy for RRMM
NCT05646836	Cev + Toc +/− XmAb24306	I	90	Open-label, Multicenter Dose-escalation Study to Evaluate the Safety, Pharmacokinetics, and Activity of XmAb24306 in Combination with Cevostamab in Patients with RRMM
NCT03275103 [146,147]	Cev +/− Toc	I	420	Open-Label, Multicenter Trial Evaluating the Safety and Pharmacokinetics of Escalating Doses of Cevostamab (BFCR4350A) in Patients with RRMM (CAMMA-3; GO39775)
NCT05535244	Cev + Toc	I/II	140	Open-Label, Multi-Cohort Study to Evaluate the Efficacy and Safety of Cevostamab in Prior B Cell Maturation Antigen-Exposed Patients with RRMM (CAMMA-2)
NCT04910568 [148]	Cev + Toc + Dex + (Dara or Pom)	I	184	Open-Label, Multicenter Trial Evaluating the Safety, Pharmacokinetics, and Activity of Cevostamab as Monotherapy and Cevostamab Plus Pomalidomide and Dexamethasone or Cevostamab Plus Daratumumab and Dexamethasone in Patients with RRMM (CAMMA-1)

Cev = cevostamab; Dara = daratumumab; Dex = dexamethasone; Pom = pomalidomide; RRMM = relapsed and/or refractory multiple myeloma; Toc = tocilizumab.

**Table 9 ijms-24-15674-t009:** Meta-analytic studies of randomized controlled trials of BCMA-targeted CAR T-cells for RRMM.

Reference	No. of Trials	Total No. of Patients	Efficacy				Toxicity	
			Pooled ORR	Pooled CRR	Pooled MRD Negativity	Median PFS (Months)	CRS ≥ Grade 3	Neurotoxicity (ICANS/CRES)
[217]	27	497	89%	13%	81%	NR	76%	8%
[218]	27	630	80.5%	71.9%	28 ^a^/68 ^b^	12.2	14.1%	20.4 ^a^/1.8 ^b^
[219]	30	950	78.3%	NR	NR	NR	6.4	3.5%
[220]	23	350	77%	37%	78%	8	14%	13%
[221]	22	681	85.2	47.0%	97.8%	14.0	6.6%	2.2%
[222]	10	353	NR	55%	NR	NR	8.0%	NR
[223]	15	285	82%	38%	77%	10	15	18
[224]	20	447	84%	36%	83%	10	15%	17%
[225]	15	285	82%	36%	77%	10	15%	18%
[226]	21	761	87%	44%	78%	8.77	11%	10%

CRES = CAR T-cell-related encephalopathy syndrome; CRR = complete response rate; CRS = cytokine release syndrome; ICANS = immune effector cell-associated neurotoxicity syndrome; MRD = minimal residual disease; ORR = overall response rate; PFS = progression-free survival. ^a^ Lymphodepletion with cyclophosphamide/fludarabine; ^b^ lymphodepletion with cyclophosphamide

## Data Availability

Not applicable.

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
