# Peer review of "Immunotherapy of Multiple Myeloma: Current Status as Prologue to the Future"

_ijms, 2023, doi:10.3390/ijms242115674_

Round 1

Reviewer 1 Report

Comments and Suggestions for Authors

Very nice review, well laid out with good structure and flow. The therapeutic modalities available for MM are well described and the reader gets a comprehensive review of the agents after reading the manuscript. 2 minor points only:

-Repetition on Paragraph 7: " high levels of cytokines at high levels". Please correct

-Moreover, CRS has been implicated as a major contributor to deadly accompanying COVID-19 infections. Please specify. Acute respiratory failure? multi-organ damage?

Comments on the Quality of English Language

 Excellent English use, no grammar or syntax issues detected.

Author Response

IJMS-2636136

Response to Reviewer 1

Comment:

Repetition on Paragraph 7: " high levels of cytokines at high levels". Please correct

Response:

This sentence has been revised to:

The first type results from T-cell recognition and activation against the targeted malignant cells followed by uncontrolled release of high levels of cytokines at high levels (on-target, on-tumor toxicity).

Comment:

Moreover, CRS has been implicated as a major contributor to deadly accompanying COVID-19 infections. Please specify. Acute respiratory failure? multi-organ damage?

Response:

This sentence has been revised to:

Moreover, CRS has been implicated as a major contributor to the acute respiratory distress syndrome and multi-organ failure deadly effects accompanying COVID-19 infections [229].

Reviewer 2 Report

Comments and Suggestions for Authors

The review is a well written summary of immune therapy of multiple myeloma including licensed medications and agents in study. It is extremely helpful to find the relevant studies by number.

It could be helpful, if the agents in question are explained in more detail with regard to immune mechanisms involved and the potential side effects including off target reactions.

Finally I am missing references to allogeneic stem cell transplantation which is the only treatment with a significant rate of cure. 

Author Response

                                                                                                                                                                                                                                                                                                                                                                                                                                                                                                                                                                                                                                                                                                                                                                                                                                                IJMS-2636136

Response to Reviewer 2

Comment: It could be helpful, if the agents in question are explained in more detail with regard to immune mechanisms involved and the potential side effects including off target reactions.

Response: Section on elotuzumab has been revised as highlighted

The signaling lymphocytic activation molecule family (SLAMF) comprises a group of transmembrane glycoproteins that is highly expressed almost exclusively on the surface of plasma cells from both normal and MM patients, as well as on natural killer (NK) cells. One member of this family, SLAMF7 (CS1), has emerged as a major target in the search for new immunological products with anti-myeloma activity [30]. The most promising of these agents, the anti-SLAMF7 humanized IgG1k mAb elotuzumab, was approved for use in combination with lenalidomide-dexamethasone in RRMM patients who had relapsed following one to three prior therapies. Elotuzumab, which unlike anti-CD38 drugs lacks single-agent activity, works through direct activation of SLAMF7 on both myeloma cells and NK cells to kill myeloma cells via ADCC. Moreover,  direct  binding of elotuzumab’s Fc portion to the FcgRIII (CD16) receptor on NK cells releases perforins and granzymes, which are cytotoxic to myeloma cells [31, 32].  Approval of elotuzumab, which unlike the anti-CD38 mAbs described above lacks single agent activity against the disease, followed analysis of results of the ELOQUENT-2 trial (NCT01239797) that included 646 patients who were randomly assigned to receive elotuzumab-dexamethasone (Ed) with or without lenalidomide (R). A 30% reduction in the risk of death or disease progression was found for the ERd cohort after one year [33]. Subsequent multi-year follow-up data confirmed the efficacy of this triplet regimen for RRMM [34-36]. Further evidence of the benefits of combining Ed treatment with an immunomodulator was provided by the ELOQUENT-3 trial (NCT02654132), which included pomalidomide in patients refractory to both lenalidomide and a proteasome inhibitor [37]. Accordingly, the elotuzumab- dexamethasone-pomalidomide regimen received FDA sanction in 2018 for the treatment of RRMM patients who had received at least two prior therapies that included these two agents. No increased safety concerns were identified in any of the studies in which elotuzumab was combined with immunomodulators. Trials combining elotuzumab- dexamethasone with the proteasome inhibitors bortezomib [38] or carfilzomib [39] in RRMM, although demonstrating significant efficacy, to date have failed to reach the level of favorable outcomes of the scale attained with elotuzumab-immunomodulator combinations. Initial efforts aimed at developing a subcutaneous dosage form of elotuzumab have been described [40].

Response: Section on elranatamab has been revised as highlighted

In Aug. 2023, the IgG2a construct elranatamab (Elrexfio®; PF- 06863135), another CD3 x BCMA bispecific that has emerged in this class, was granted accelerated approval as monotherapy by the FDA [107]. This action was spurred by the results from the MagnetisMM-3 trial (NCT04649359) in which an ORR of 61.0% and a complete response rate of 35.0% were reported in a cohort of 123 RRMM subjects after a median follow-up of 14.7 months [108]. The most frequently observed Grade 3 or higher adverse events in this trial were infections (39.8%), anemia (37.4%), and neutropenia (48.8%). CRS of any grade was found to occur in 57.7% but no patients experienced CRS higher than grade 2. Overall, biweekly dosing was found to reduce total grade 3 or 4 adverse events from 58.6% to 46.6%, indicating that biweekly dosing may offer the advantage of improved safety without compromising efficacy [108].  Table 4 contains a listing of several ongoing clinical trials in which elranatamab is under evaluation.

Common adverse events (any grade; grade 3–4) included infections (69.9%, 39.8%), cytokine release syndrome (57.7%, 0%), anemia (48.8%, 37.4%), and neutropenia (48.8%, 48.8%). With biweekly dosing, grade 3–4 adverse events decreased from 58.6% to 46.6%. Elranatamab induced deep and durable responses with a manageable safety profile. Switching to biweekly dosing may improve long-term safety without compromising efficacy.

Response: Section on talquetamab has been revised as highlighted

In Aug. 2023, the FDA granted accelerated approval to talquetamab (Talvey® by Janssen) a GPRC5D x CD3 bispecific [133], based on the results of the MonumenTAL-1 trial (NCT03399799, NCT4634552) in which ORRs of 73.0% and 73.6% were obtained in respective cohorts receiving subcutaneous doses of either 0.4 mg/kg weekly (N = 100) or 0.8 mg/kg biweekly (N = 87). The most common adverse events attributed to talquetamab were mainly low grade and consisted of CRS, altered taste, and skin-related.  Table 6 shows the current ongoing trials of talquetamab in RRMM patients.

Other sections already describe the underlying mechanisms for each immunotherapy.

Comment: Finally I am missing references to allogeneic stem cell transplantation which is the only treatment with a significant rate of cure.

Response: The following highlighted paragraph has been added to the Summary and Conclusions Section:

         Finally, although ASCT has been a standard of care for transplant-eligible MM patients since its introduction nearly forty years ago, its hematopoietic source counterpart, namely allogeneic stem cell transplantation (allo-SCT) has received comparatively little attention until rather recently. Allo-SCT, which has been described as having the potential to yield curative outcomes in the disease, is based on the graft vs. myeloma effect first described in 1996 [268].  While at present this modality is generally reserved for young patients having high-risk relapsed myeloma, it remains controversial due to its overall mixed efficacy and lack of clear treatment guidelines [269]. On the other hand, this approach to MM therapy bears watching for future developments and possibly wider applications [270].

Reviewer 3 Report

Comments and Suggestions for Authors

1. It is a good and comprehensive summary of immunotherapies on multiple myeloma.

2. In each section, adding the underlying mechanisms /signaling pathways of each immunotherapy/each molecular in curing the disease would be good.

3.   Adding the information of those medicine companies would be good.

Comments on the Quality of English Language

English is good.

Author Response

IJMS-2636136

Responses to Reviewer 3

Comment: In each section, adding the underlying mechanisms /signaling pathways of each immunotherapy/each molecular in curing the disease would be good.

Response: Section on elotuzumab has been revised as highlighted

The signaling lymphocytic activation molecule family (SLAMF) comprises a group of transmembrane glycoproteins that is highly expressed almost exclusively on the surface of plasma cells from both normal and MM patients, as well as on natural killer (NK) cells. One member of this family, SLAMF7 (CS1), has emerged as a major target in the search for new immunological products with anti-myeloma activity [30]. The most promising of these agents, the anti-SLAMF7 humanized IgG1k mAb elotuzumab, was approved for use in combination with lenalidomide-dexamethasone in RRMM patients who had relapsed following one to three prior therapies. Elotuzumab, which unlike anti-CD38 drugs lacks single-agent activity, works through direct activation of SLAMF7 on both myeloma cells and NK cells to kill myeloma cells via ADCC. Moreover,  direct  binding of elotuzumab’s Fc portion to the FcgRIII (CD16) receptor on NK cells releases perforins and granzymes, which are cytotoxic to myeloma cells [31, 32].  Approval of elotuzumab, which unlike the anti-CD38 mAbs described above lacks single agent activity against the disease, followed analysis of results of the ELOQUENT-2 trial (NCT01239797) that included 646 patients who were randomly assigned to receive elotuzumab-dexamethasone (Ed) with or without lenalidomide (R). A 30% reduction in the risk of death or disease progression was found for the ERd cohort after one year [33]. Subsequent multi-year follow-up data confirmed the efficacy of this triplet regimen for RRMM [34-36]. Further evidence of the benefits of combining Ed treatment with an immunomodulator was provided by the ELOQUENT-3 trial (NCT02654132), which included pomalidomide in patients refractory to both lenalidomide and a proteasome inhibitor [37]. Accordingly, the elotuzumab- dexamethasone-pomalidomide regimen received FDA sanction in 2018 for the treatment of RRMM patients who had received at least two prior therapies that included these two agents.

Other sections already describe the underlying mechanisms for each immunotherapy.

Comment:

Adding the information of those medicine companies would be good.

Response:

The trade/brand name and manufacturer has been included with the accepted generic name of each drug that has been approved for clinical use. Code designations have been deleted in these cases but have been retained for drugs that have not yet been approved.  Changes made shown below:

daratumumab add (Darzalex by Janssen)

Isatuximab (SAR650984),  add (Sarclisa by Sanofi)

Elotuzumab add (Empliciti by Bristol-Myers Squibb)

belantamab mafodotin (Blenrep®, belamaf, GSK2857916) by   GSK)

teclistamab (Tecvayli® by Janssen)

elranatamab (Elrexfio®; PF- 06863135) by Pfizer)

talquetamab (Talvey®) by Janssen)

Bluebirdbio’s idecabtagene vicleucel (ide-cel; Abecma®; bb2121; ide-cel) by Bristol-Myers Squibb)

ciltacabtagene autoleucel (Carvykti®, JNJ- 68284528; LCAR-B38M; cilta-cel) by Janssen and Legend